# Zebularine, a DNA Methylation Inhibitor, Activates Anthocyanin Accumulation in Grapevine Cells

**DOI:** 10.3390/genes13071256

**Published:** 2022-07-15

**Authors:** Junhua Kong, Virginie Garcia, Enric Zehraoui, Linda Stammitti, Ghislaine Hilbert, Christel Renaud, Stéphane Maury, Alain Delaunay, Stéphanie Cluzet, Fatma Lecourieux, David Lecourieux, Emeline Teyssier, Philippe Gallusci

**Affiliations:** 1UMR Ecophysiologie et Génomique Fonctionnelle de la Vigne, Université de Bordeaux, INRAE, Bordeaux Science Agro, 210 Chemin de Leysotte—33140 Villenave d’Ornon, France; kongjunhua@ibcas.ac.cn (J.K.); virginie.garcia@inrae.fr (V.G.); enric.zehraoui@inrae.fr (E.Z.); linda.stammitti@inrae.fr (L.S.); ghislaine.hilbert@inrae.fr (G.H.); christel.renaud@inrae.fr (C.R.); fatma.ouaked-lecourieux@inrae.fr (F.L.); david.lecourieux@inrae.fr (D.L.); philippe.gallusci@inrae.fr (P.G.); 2INRAe, EA1207 USC1328 Laboratoire de Biologie des Ligneux et des Grandes Cultures, Université d’Orléans, 45067 Orléans, France; stephane.maury@univ-orleans.fr (S.M.); alain.delaunay@univ-orleans.fr (A.D.); 3Unité de Recherche Oenologie, Faculté des Sciences Pharmaceutiques, University Bordeaux, EA4577, USC 1366 INRA, Equipe Molécules d’Intérêt Biologique (GESVAB), ISVV, CEDEX, 33882 Villenave d’Ornon, France; stephanie.cluzet@u-bordeaux.fr

**Keywords:** DNA methylation, zebularine, anthocyanin

## Abstract

Through its role in the regulation of gene expression, DNA methylation can participate in the control of specialized metabolite production. We have investigated the link between DNA methylation and anthocyanin accumulation in grapevine using the hypomethylating drug, zebularine and Gamay Teinturier cell suspensions. In this model, zebularine increased anthocyanin accumulation in the light, and induced its production in the dark. To unravel the underlying mechanisms, cell transcriptome, metabolic content, and DNA methylation were analyzed. The up-regulation of stress-related genes, as well as a decrease in cell viability, revealed that zebularine affected cell integrity. Concomitantly, the global DNA methylation level was only slightly decreased in the light and not modified in the dark. However, locus-specific analyses demonstrated a decrease in DNA methylation at a few selected loci, including a CACTA DNA transposon and a small region upstream from the *UFGT* gene, coding for the UDP glucose:flavonoid-3-*O*-glucosyltransferase, known to be critical for anthocyanin biosynthesis. Moreover, this decrease was correlated with an increase in *UFGT* expression and in anthocyanin content. In conclusion, our data suggest that *UFGT* expression could be regulated through DNA methylation in Gamay Teinturier, although the functional link between changes in DNA methylation and *UFGT* transcription still needs to be demonstrated.

## 1. Introduction

Anthocyanins are colored flavonoids accumulating in the vacuoles of cells from diverse plant tissues. Because they contribute to flower and fruit colors, as for example in grape berries, they are responsible for the attraction of pollinators and herbivores, therefore facilitating pollen and seed dispersal. In addition, anthocyanins have been associated with photoprotection and free radical scavenging [1]. It is also speculated that they contribute to stress tolerance since their synthesis is up-regulated in response to many different abiotic stresses including drought, salinity, excess light, sub- or supra-optimal temperatures, and nitrogen and phosphorus deficiency [2]. In addition, anthocyanins have lately emerged as components of high economical interest since they are used as natural colorants with healthful properties [3]. Anthocyanin synthesis has been characterized in a number of species, revealing very well conserved features. Briefly, anthocyanin biosynthesis is divided into two main parts: phenylalanine is first converted to *p*-coumaroyl-CoA through the phenylpropanoid pathway. The flavonoid pathway is then initiated by the coupling of *p*-coumaroyl-CoA with malonyl-CoA, leading to the production of flavonols, proanthocyanidins, and anthocyanins. Anthocyanins ultimately derive from the unstable anthocyanidins by glycosylation and eventually acylation (Figure 1) [4,5]. The genes coding for the different biosynthetic enzymes and vacuolar transporters have been identified in grapevine (Appendix A, [6,7,8,9,10,11,12,13,14,15,16,17,18,19,20,21,22,23,24]), many of them belonging to multigene families, especially those corresponding to early steps of the biosynthesis pathway [4]. Interestingly, gene expression analyses have revealed that the induction or repression of anthocyanin biosynthesis is primarily regulated at the transcriptional level [4,5]. In particular, the expression of UFGT (UDP glucose:flavonoid-3-*O*-glucosyltransferase), which is responsible for the conversion of anthocyanidins in anthocyanins, was shown to be critical for anthocyanin biosynthesis [25]. As a matter of fact, a complex network involving different genes coding for MYB, basic helix–loop–helix and WD40 repeat transcription factors has been shown to control anthocyanin biosynthesis (Appendix A, [23,26,27,28,29,30,31,32,33,34]). Among the transcription factors, MYBA1 plays a central role, as an inducer of UGFT transcription. Finally, different reports suggest that anthocyanin biosynthesis is regulated through DNA methylation in a number of plant species.

DNA methylation corresponds to the addition of methyl groups at the C5 carbon of cytosine. In plants, cytosines can be methylated in any sequence contexts (CG CHG and CHH, where H stands for any nucleotide except G). High cytosine methylation levels in all contexts are associated with transposon silencing so that DNA methylation is often described as a protective mechanism for genomic DNA. However, a number of studies have also revealed a correlation between the methylation status of gene promoters and the gene expression level. In most cases, a high methylation level at the promoter in all sequence contexts is associated with a reduced gene expression [35], although in a few cases the reverse situation was demonstrated, as for example for the *Arabidopsis ROS1* gene [36,37]. Moreover, transcriptional regulation through DNA methylation has been shown to play important roles during plant development and in response to environmental changes [38,39,40,41]. Interestingly, the genes whose expression regulation involves DNA methylation include a number of genes related to plant metabolism and in particular to plant secondary metabolism. As an example, carotenoid accumulation in tomato fruit during the ripening process was shown to be correlated with the demethylation of the promoter of the PHYTOENE SYNTHASE gene [42], together with the demethylation of central regulators of tomato fruit ripening, such as RIN, NOR, and CNR [42,43]. Moreover, these demethylation events are required for the ripening to be initiated in the tomato fruit [42]. As shown in this study the regulation of gene expression through DNA methylation may concern both structural genes coding for enzymes and regulatory genes coding for transcription factors.

Concerning the regulation of anthocyanin biosynthesis, several studies proposed that the methylation status of *MYB* regulator genes could play a critical role based on the characterization of natural anthocyanin-deficient variants of apple [44,45], pear [46], and radish [47]. In these studies, the decrease in anthocyanin production in the variants was shown to be correlated with the silencing of a key *MYB* regulatory gene, concomitantly with an increase in its promoter methylation status. Of particular interest is the study performed by Wang et al. (2013) in pear demonstrating that *PcMYB10* transcriptional gene silencing through targeted methylation by VIGS could phenocopy the anthocyanin-deficient variant [46]. Inversely in a recent study performed in radish, Wang et al. treated the white variant using the hypomethylating drug, 5-azacytidine, and were able to induce the reactivation of *RsMYB1* concomitantly with the partial reversal of the white phenotype [47]. A few reports also described a correlation between changes in the anthocyanin content, *MYB* gene expression and the methylation status of *MYB* promoter in response to environmental changes [48,49] and/or during the development [50]. Of note, genes other than *MYB* were also shown to be associated with changes in DNA methylation concomitantly with variations in their expression level and with modification in anthocyanin concentration [51,52]. For example, Liu et al. (2012) demonstrated that the lack of anthocyanin pigmentation in the perianth of a natural variant of *Oncidium* orchids was correlated to the very low expression of the chalcone synthase encoding gene (*ogCHS*), whereas no difference could be detected in the expression of different regulator genes. Moreover, a methylation assay suggested that the methylation status of the 5’ upstream DNA region of *ogCHS* was more methylated in the non-pigmented variant [51]. Altogether these studies suggest that the biosynthesis of anthocyanin could be regulated through DNA methylation.

In order to analyze the functional relationship between DNA methylation and anthocyanin biosynthesis in grapevine cells, we used zebularine, an already characterized DNA methylation inhibitor [53,54], as an alternative to transgenesis since the production of transgenic grapevine plants is still challenging. Zebularine is a non-methylable structural analog of cytidine, which is incorporated into genomic DNA. Different studies using animal models have revealed that the interaction between DNA methyltransferases and zebularine produce stable complexes leading to DNA methyltransferases trapping and inactivation in the form of covalent protein–DNA adducts [55,56]. As a result, DNA methyltransferases are rapidly depleted, and genomic DNA becomes demethylated when DNA replication occurs [55,56]. Genome-wide analyses in *Arabidopsis thaliana* suggested that the hypomethylating effect of zebularine affects the entire genome in all sequence contexts, targeting similarly different types of methylated regions including, gene body methylated genes, genes methylated in the CHH context and transposons [54]. Interestingly, several studies focusing on specific loci have shown that the hypomethylating effect of zebularine was correlated with the upregulation of sequences normally submitted to silencing including an endogenous imprinted gene [53,54], a transgene [57], or a transposon [58]. Compared to other DNA methyltransferase inhibitors, zebularine is more stable and earlier studies also suggested that its toxicity was limited [55,59]. However, it now becomes clear that zebularine, as well as azacytidine, induce DNA damages leading to cell division arrest and the activation of DNA damage repair (DDR) pathways [60,61,62,63]. This observation probably explains why zebularine treatments have been repeatedly associated with reduced plant growth [53,58,64,65,66].

The present work aimed at investigating the potential role of DNA methylation in regulating anthocyanin biosynthesis in grapevine, taking advantage of the grapevine Gamay Teinturier (GT) cells, which naturally produce anthocyanins when grown under light but do not in the dark. Data indicate that anthocyanin accumulation is stimulated or induced depending on the light conditions when zebularine is added to the growth medium. In an attempt to clarify the mechanisms underlying these observations, the elicited cell suspensions have been extensively characterized using complementary approaches including anthocyanin quantification, microscopic observations, DNA methylation analyses, and transcriptomics. Despite the very limited effects of zebularine on global grape cell DNA methylation levels, we provide evidence that zebularine treatments induce a slight decrease in the methylation level at the *UFGT* gene promoter region, which correlates with an induction/upregulation of its expression. In addition, the analysis of the mRNA populations revealed that zebularine has a strong impact on cell physiology and may generate a stress response indirectly involved in the accumulation of anthocyanins.

## 2. Materials and Methods

### 2.1. Plant Material

*Vitis vinifera* (L.) cv. “Gamay Fréaux” var. Teinturier (*Vitaceae*) cell suspensions were established from berries as described previously [67]. GT cell suspension cultures were initiated from fresh friable calli in a modified MS liquid medium (M0221, Duchefa, Haarlem, The Netherlands) supplemented with 20 g/L sucrose (S0809, Duchefa), 0.25 g/L N-Z-Amine A (C7290, Sigma, St. Louis, MO, USA), 0.5 mg/L auxin, 0.1 mg/L cytokinin and vitamins (100 mg/L myo-inositol, 1.0 mg/L nicotinic acid, 1.0 mg/L pantothenic acid, 0.01 mg/L biotin, 1.0 mg/L pyridoxine HCl, and 1.0 mg/L thiamine HCl) [67]. Cells were subcultured in 50 mL MS liquid medium with a 1/5 (*v*/*v*) ratio every 12 days using 250 mL Erlenmeyer flasks and maintained at 25 °C (+/−1 °C) with constant shaking (120 rpm) under continuous fluorescent light (5000 lux) or at 24 (+/−1) °C in dark conditions (light off and Erlenmeyer flasks wrapped with aluminum foil).

Preliminary experiments were conducted using two different inhibitors of DNA methyltransferases (DMTs), with distinct modes of action, zebularine and RG108. Zebularine is a structural analog of methylcytosine (mC), which is incorporated into the DNA molecules where it can establish stable bounds with DMT and inactivate them [55,56]. In contrast, RG108, which is not a structural analog of mC, was isolated through its ability to bind to and block the active site of DMT [68]. It has been only slightly characterized in animal systems, and to our knowledge it has never been used in plants. Treatments with the DNA methyltransferase inhibitors zebularine (Selleckchem, Houston, TX, USA, S7113), or RG108 (Cliniscience, Nanterre, France, HY-13642) were performed the 3rd day after subculture at the end of the lag phase. Alternatively, DMSO, which was used as a solvent for zebularine and RG108, was added to the medium. An additional control with water was performed. Cells were harvested by vacuum filtration, quickly washed twice with MS medium and immediately frozen in liquid nitrogen. Frozen grape cell samples were ground into fine powder in liquid nitrogen and stored at −80 °C for further analyses.

### 2.2. Anthocyanin, Sugar and Amino Acid Quantification

Anthocyanins were extracted and analyzed using freeze-dried powders prepared from cell samples, essentially as described in [69]. Briefly, 20 mg of freeze-dried powder were resuspended in 300 μL of methanol (0.1% HCl) to extract anthocyanin before filtering through a 0.2 µM syringe filters, before injecting 3 μL for HPLC analysis. The integrated absorbance at 520 nm was used to determine the concentration of individual anthocyanin expressed as malvidin 3-glucoside equivalents (Extrasynthese, Genay, France) calculated from a calibration function obtained on the commercial standard.

Sugars and amino acids were extracted and analyzed as described in [70].

### 2.3. Stilbene Quantification 

Stilbene extraction was performed from freeze-dried cells (10–20 mg DW) overnight at 4 °C with 3 mL methanol (MeOH). After centrifugation (5 min) at 3500 rpm, 2 mL of supernatant were recovered, and a second extraction was carried out with 3 mL MeOH for 1 h 30 at room temperature. Tubes were centrifuged as previously described and 3 mL of supernatant were recovered. Cells were extracted a third time with 3 mL MeOH during 1 h 30 and 3 mL of supernatant were recovered. The three recovered supernatants were pooled and speed-vacuum evaporated. Dried extract was resuspended in 500 µL MeOH/H_2_O (50/50, *v*/*v*) and filtered (0.45 µm PTFE) before HPLC analysis. The analysis of stilbene content was performed by HPLC (Agilent 1100 Series, Agilent Technologies, Santa Clara, CA, USA), on a 250 mm × 4 mm Prontosil C18 (5 µm) reverse-phase C18 column (Bischoff Chromatography, Leonberg, Germany) protected by a guard column of the same material. Separation was performed at a flow rate of 1mL/min with a mobile phase composed of (A) H_2_O: TFA 1% (97.5/2.5, *v*/*v*) and (B) Acetonitrile: A (80/20, *v*/*v*). The run was set as follows: 0–1 min, 20% (B); 1–8 min, from 20% (B) to 24% (B); 8–10 min, from 24% (B) to 25% (B); 10–13 min, 25% (B); 13–18 min, from 25% (B) to 30% (B); 18–35 min, from 30% (B) to 50% (B); 35–37 min, from 50% (B) to 100% (B); 37–41 min, 100% (B); 41–42 min, from 100% (B) to 20% (B); and 20% (B) for 4 min. The chromatogram was monitored at 286 and 306 nm using a UV–visible-DAD detector (Agilent Technologies, Santa Clara, CA, USA). Stilbene (*trans*-resveratrol, *trans*- and *cis*-piceids) contents (in µg.g^−1^ FW) were determined from calibration curves of pure standards (injected concentrations ranging from 2 to 500 μg/mL). *Trans*-Resveratrol was purchased from Sigma Chemical Co. (St. Louis, MO, USA). *Cis*- and *trans*- Piceids were purified from *V. vinifera* L. cv GT cell cultures as described by Waffo Teguo et al. [71]. The linearity of the response of the standard molecules was checked by plotting the peak area versus the concentration of the compounds.

### 2.4. Nucleic Acid Extraction

Total RNAs were isolated as described in [72], before the elimination of genomic DNA with Dnase I (Turbo DNA-free TM kit, Ambion, Austin, TX, USA) according to the manufacturer instructions. For genomic DNA preparation, 5 mL of pre-warmed (65 °C) extraction buffer [Tris HCl (pH 8.0) 100 mM, EDTA 20 mM, NaCl 1.4 M, CTAB 4%, PVPP 1%, β-mercaptoethanol 0.5%] were added to 500 mg cell powder. The samples were thoroughly mixed and incubated at 65 °C for 1 h with regular shaking. After two chloroform:isoamyl alcohol (24:1) extractions, the aqueous phase was recovered and centrifuged at 20,000× g for 15 min at room temperature. Genomic DNA was then precipitated with 0.5V ammonium acetate 7.5 M and 2.5V cold absolute ethanol. After 2 h at −20 °C, the nucleic acids were collected by centrifugation at 20,000× *g* for 30 min at room temperature, washed twice with pre-cooled 70% ethanol, air dried, and dissolved in 400 μL TE buffer (pH 8.0). Rnase A was added at 20 µg/mL and the solution was incubated 30 min at 37 °C. Nucleic acid concentrations and 260/280 nm ratios were determined with a Nano-Drop 2000c spectrophotometer (Thermo Fisher Scientific, Wilmington, DE, USA).

### 2.5. RT-qPCR Analysis

Retrotranscription (RT) was performed with 1 µg of total RNA using the iScript cDNA synthesis kit (Bio Rad, Hercules, CA, USA, #1708891), according to the manufacturer’s instructions. Each RT reaction was completed in duplicate. Real-Time RT-PCR analysis was performed using Biorad C1000 Touch or Biorad CFX Connect thermocyclers and CFX Real-Time PCR detection systems, according to the manufacturer’s instructions. PCR efficiency was measured for each primer pair on cDNA standards. PCR reactions were performed in 96-well plate using the iTaq™ Universal SYBR^®^ Green Supermix kit (Biorad #1725124), with 5 pmol of each primer and 1 µL of ten-fold diluted RT reaction in a final volume of 10 µL. Reactions were run using the manufacturer’s recommended cycling parameters (95 °C for 3 min, 40 cycles of 95 °C for 15 s and 60 °C for 30 s). No-template controls were included for each primer pair and each PCR reaction was completed in duplicate. Melting curves were analyzed to verify the specificity of each amplification reaction. The primers used for the qPCR reactions are indicated in Appendix A. The amplification of EF1α was used as a control for normalization. For each gene, differences between samples were calculated using the ΔΔCt method.

### 2.6. Methylation Analysis with HPLC and McrBC-PCR

Genomic DNA (5 μg) enzymatically hydrolyzed into nucleosides was analyzed by high-performance liquid chromatography (HPLC) following a previously published protocol and control procedure [73,74]. The methylcytosine percentages (%mC) were calculated as follows: %mC = (mC/(C + mC)) × 100, where C represents 2-deoxycytidine content and mC represents 5-methyl-2-deoxycytidine content. Genomic DNA (500 ng) was digested with 50U of McrBC (New England Biolabs, Ipswich, MA, USA, #M0272) for 5 h at 37 °C in a final volume of 50 µL, according to the manufacturer’s instructions. After enzyme inactivation, real time PCR reactions were performed as described above using 1 µL of McrBC reaction in a final volume of 10 µL. Amplification of actin (unmethylated locus) was used as a control for normalization. The primers used for the McrBC-PCR reactions are indicated in Appendix A. For each targeted genomic region, differences between the zebularine-treated samples (or the water-control samples) and the DMSO-control samples were calculated using the ΔΔCt method as described in [72,75].

### 2.7. RNA-Seq Analysis

The paired-end reads were cleaned and trimmed with Trimmomatic [76] version 0.38 (with the options PE, LEADING:3, TRAILING:3, SLIDINGWINDOW:4:15 and MINLEN:36). Hisat2 (version 2.2.0) (Kim, Langmead, & Salzberg, 2016) with default parameters was used to align filtered reads to the 12X.v2 version of the grapevine reference genome sequence from the French-Italian Public Consortium (PN40024) with the associated structural annotation (Vcost.v3) provided by URGI (https://urgi.versailles.inra.fr/Species/Vitis/Data-Sequences/Genome-sequences (accessed on 12 October 2018)). The count matrices were created by directly importing BAM alignments in DESeq2 [76,77] (R version 3.5.1, DESeq2 version 1.22.2) as well as the gene models described in the previously used gff file. Reads *per* gene were counted with the summarize Overlaps function with “Union” mode and transformed with the rlog function. Sample-to-sample distances were visualized with PCA plots calculated by the plotPCA function provided by DESeq2 package on the rlog transformed values. Differential gene expression analysis was carried out with the DESeq2 pipeline. All the contrasts of interest were extracted from the results and only items with an adjusted *p*-value < 0.05 and a|log2FC| > 1.0 were selected for downstream analysis. Genes with low expression levels (RPKM < 1 in all samples) were eliminated. Gene Ontology enrichment analysis of differentially expressed genes was carried out with the topGO package (topGO version 1.0, R version 3.6.1) and GO terms with corrected *p*-value less than 0.05 were considered significantly enriched. They were represented with the protocol for informative visualization of enriched Gene Ontology terms [78]. All the gene models were automatically categorized according to the MapMan ontology (version 3.6) with Mercator tool [79] and MAPMAN standalone software (v3.5.1) was used to explore the data.

## 3. Results

Gamay Teinturier (GT) cell suspensions were used to evaluate the possible role of DNA methylation in anthocyanin biosynthesis in grapevine. These cells, which derive from the pulp of GT berries, strictly require light to accumulate anthocyanins [80]. Preliminary experiments suggested that two different drugs, zebularine and RG108, both inhibiting DNA methylation, stimulate anthocyanin biosynthesis in light grown GT cells, as revealed by the cell suspension color, and by anthocyanin quantification (Appendix A). Furthermore, both drugs appeared to induce anthocyanin biosynthesis in dark grown GT cells (Appendix A). These results suggested that DNA methylation may play a role in the regulation of anthocyanin biosynthesis in GT grape cells. They prompted us to initiate a more comprehensive analysis, including the description of the cell metabolic and transcriptomic states after the drug treatment. Zebularine was chosen to perform this study, for its better characterization in the literature.

### 3.1. Zebularine Stimulates Anthocyanin Production

#### 3.1.1. Zebularine Inhibits Cell Growth and Impacts Cell Color

Zebularine was added to GT cells 3 days after subculture at the beginning of the exponential growth phase (Figure 2A,B), in order to ensure an efficient incorporation of zebularine into the DNA molecules. In addition, cells cultured with the addition of water or of DMSO (zebularine solvent), instead of zebularine were used as controls. The zebularine treatment had a strong inhibitory effect on GT cell growth and induced a significant change in color. Inhibition of cell growth happened in the presence and absence of light in a dose dependent manner. More precisely, the addition of 25, 50, or 75 µM zebularine resulted, after 9 days in culture, in a 30, 40, and 46% reduction in FW accumulation, for cells grown in light and in a 34, 45, and 50% reduction for those grown in the dark (Figure 2A,B), as compared to control conditions. However, zebularine effects were visible as early as 3 to 4 days after the drug supplementation depending on the light/dark conditions.

Concerning cell color, cells were differently affected depending on the light conditions. In light, control cells already accumulate anthocyanins [3,81] (Figure 2C). However, the color of zebularine-treated cells appeared more intense than the one of the control cells (Figure 2C), consistent with a stimulatory effect of the zebularine treatment on anthocyanin synthesis. In the absence of light, GT cells do not accumulate anthocyanin [80], which was confirmed in our conditions since dark grown control GT cells remained uncolored after 9 days in culture (Figure 2D). In contrast, zebularine-treated cells appeared pink, suggesting that a zebularine treatment was sufficient to induce anthocyanin production in the absence of light (Figure 2D).

Cells were harvested 12 days after subculturing, after a 9 day long zebularine treatment, and subjected to multiple analyses, in order to characterize the drug effect.

#### 3.1.2. Zebularine Treatment Results in an Increase in the Proportion of Colored Cells

The microscopic observation of control light grown cells 12 days after subculturing revealed their heterogeneity: only a subset of cells effectively produced anthocyanins, with the vast majority of cells remaining colorless or pale pink (Figure 3A), as was already reported for GT cell suspensions [82,83]. Cell culture heterogeneity was still observed in the presence of zebularine, although the proportion of colored cells increases (Figure 3A). In the dark, zebularine treatments induced the appearance of colored cells with variable color intensities, often grouped as small clusters among a majority of cells which stay colorless (Figure 3B). This suggests that in both situations, light and dark, zebularine elicited a limited number of cells.

#### 3.1.3. Zebularine Treatments Induce an Increase in Anthocyanin Quantities

In the light conditions, 12 days after subculturing, water and DMSO control cells accumulated approximately the same amount of total anthocyanins, close to 4.7 mg/g DW (Figure 4), which was enhanced 1.3-, 1.6-, and 2.2-fold in the presence of 25, 50, and 75 µM zebularine, respectively (Figure 4). In control cells, whether they were supplemented with water or DMSO, as already described for light grown GT cell suspensions [84], the two dihydroxylated anthocyanins, peonidins, and cyanidins, represented more than 90% of the total anthocyanin content, with 76.6% peonidins and 16.9% cyanidins (Figure 4). Tri-hydroxylated forms, i.e., malvidin, petunidin and delphinidin, were also detected (Figure 4), but their quantity represented only 4.7, 1.1, and 0.8% of the total anthocyanin content (Appendix A). The relative proportions of the different anthocyanins were not dramatically impacted by zebularine treatments (Figure 4 and Appendix A), even when considering the various chemical modifications (glycosylation, coumaroylation, and acetylation) (Appendix A). Peonidin remained by far the most abundant anthocyanin, although its relative abundance decreased from 77% in DMSO-treated cells to 69, 71, and 72% in cells treated with 25, 50, and 75 µM of zebularine, respectively (Figure 4). This decrease correlates with a concomitant increase in tri-hydroxylated anthocyanins, from 6% in DMSO-treated cells to 14, 10, and 9% in cells treated with zebularine 25, 50, and 75 µM, respectively, whereas the proportion of cyanidin remained unchanged (Figure 4).

It is well described that GT cells do not accumulate anthocyanin in the absence of light [3]. In our dark growing conditions, both water- and DMSO- treated cells produced a trace amount of anthocyanins (below 40 μg/g DW). However, zebularine treatments resulted in a dose-dependent increase in the total anthocyanin accumulation, reaching 412, 635, and 1020 µg/ g DW in 25, 50, and 75 µM zebularine-treated samples, respectively (Figure 4). Of note, the quantity of anthocyanins accumulating in the zebularine-treated dark grown cells remains much lower than the amount detected in control light grown cells (Figure 4). Interestingly, only cyanidin and peonidin derivatives, the two most abundant compounds produced in light cultured cells were detected in dark grown cells (Figure 4), suggesting that light is required for the accumulation of the trihydroxylated compounds.

### 3.2. Transcriptome Analysis Suggests Complex Effects of Zebularine on Gene Expression in GT Cells

In order to decipher the mechanisms underlying the effect of zebularine on the physiology of GT cells, a transcriptomic analysis was performed using the exact same samples as those used for anthocyanin quantification. Between 9.5 and 16.8 million raw reads were generated for each sample, leading to 9.0 to 16.1 million reads after filtering. Between 93.9 and 95.8% of these reads were mapped to the 12X.v2 grape reference genome [85]. Finally, a total of 21,955 genes were expressed in at least one light sample and 22,010 genes in at least one dark sample, representing approximately 83% and 84% of all identified genes in the grape genome with Vcost.v3 annotation. Taking into consideration the five selected conditions (Light DMSO (LD), light zebularine 50 µM (LZ50), dark DMSO (DD), dark zebularine 50 µM (DZ50), and dark zebularine 25 µM (DZ25)), all possible pairwise comparisons of RNA-SEQ results were made in order to determine the differentially expressed genes (DEGs). A principal component analysis (PCA) using DEGs from all DMSO and zebularine-treated samples showed a clear separation along the PC1 axis, which accounts for 46.51% of the variance of cells grown in control conditions depending on the light regime. In contrast, zebularine-treated samples grouped together suggesting that the impact of light on the GT cell transcriptome was in part compensated in the presence of zebularine (Figure 5A). Cells grown in the dark behaved similarly irrespective of the zebularine concentration used, 25 µM (DZ25) or 50 µM (DZ50). As a consequence, for clarity reasons, only the samples treated with 50 µM of zebularine were considered for the following analysis. As expected from the PCA analysis, the highest number of DEGs was found for the LD/DD comparison with 6282 DEGs (Figure 5B), representing 83% of all DEGs, including 3456 down-regulated and 2826 up-regulated genes.

In contrast, the comparison of LZ50 and DZ50 revealed a much lower number of DEGs (1166 DEGs), corroborating the PCA analysis, consistent with a convergence of light and dark grown cell transcriptomes after zebularine treatments. Whereas, in light, zebularine alters the expression of 4496 genes, far fewer genes were identified as zebularine-modulated in the dark (1209 DEGs). Interestingly, each pairwise comparison revealed both up-regulated and down-regulated genes (Figure 5B). As shown in the Venn diagram presented in Appendix A, many DEGs are shared between two or more comparisons. In particular, 80% of the genes deregulated by zebularine in the light (LZ50 vs LD) were also identified as differentially regulated by light in the control conditions (LD vs DD). As described below, the concordance between these two sets of genes was also revealed by a GO analysis and MAPMAN [86] overviews of primary and secondary metabolic pathways.

The genes up-regulated by light in the control cells were enriched in 20 different GO terms mostly linked to metabolic functions (Figure 6A). In addition to photosynthesis which was the most significant GO term (adjusted *p*-value < 10^−30^), others include mannose, trehalose, fructose and serine family amino acid metabolic processes, carbon utilization, and pentose-phosphate shunt. Genes down-regulated by light in control cells were enriched in 53 different GO terms (Figure 6C). Interestingly, 23 of these GO functions were related to DNA replication, mitotic (or meiotic) cell division, microtubule organization, and cell wall reorganization, revealing that in the light, there was a global repression of genes linked to the cell cycle progression in the control cells, by comparison with the dark condition. The other functions enriched among the light down-regulated DEGs include ribosome biogenesis, translation, as well as 16 different metabolic processes. In order to get a better understanding of the metabolic deregulations occurring in the control cells (LD versus DD), a MAPMAN analysis was conducted (Appendix A). The representation of the primary and secondary metabolic pathways showed a global repression of the transcription of genes related to the central metabolism (mitochondrial electron transport, tricarboxylic acid cycle), with the notable exception of photosynthesis-related genes which were induced in light grown cells (Appendix A). Other deregulated pathways include amino acid degradation and synthesis, as well as secondary metabolism pathways, and minor CHO metabolism. For example, two genes coding for myo-inositol oxygenases (Vitvi09g00246, MIOX2 and Vitvi11g00231, MIOX4) were detected with 21- and 13-fold induction in the light DMSO-treated cells compared to the dark DMSO-treated cells. In Arabidopsis, expression of the MIOX2 and MIOX4 genes was correlated with low energy/nutrient conditions [87,88]. Altogether the GO and MAPMAN analyses revealed characteristics that are reminiscent of a carbon deficiency response, suggesting that in the light, 12 days after sub-culturing, cells face carbon starvation. These characteristics were not detectable in the other cell samples, suggesting that the cells grown in the dark or in the presence of zebularine do not suffer from carbon starvation or to a lesser extent. Strikingly the GO analysis of the DEGs up-regulated by zebularine in the light led to the identification of 30 different functions, 28 of which were identical to the functions associated with the DEGs up-regulated in the dark for the control cells (Figure 6C,D). Similarly, the genes that were down-regulated by zebularine in the light correspond to the same GO categories as the genes down-regulated in DD cells compared to LD cells (Figure 6A,B). Moreover, MAPMAN representations of the metabolism-related gene deregulation revealed two very similar pictures for the two comparisons LD/DD and LD/LZ50 (Appendix A).

We also examined the GO enrichment of DEGs down-regulated by zebularine in the dark. Genes involved in carbohydrate metabolism, cell wall modification, cytoskeleton reorganization, as well as stress responses (hexachlorocyclohexane metabolic process, abscisic acid activated signaling pathway, hydrogen peroxide catabolic process, and response to oxidative stress) were the most represented (Figure 7). Furthermore, a MAPMAN analysis suggests a global repression of the central metabolism (Appendix A). Finally, only 2 GOs are detected as specifically enriched among the 245 up-regulated genes: transmembrane transport and oxidation–reduction process (Figure 7). No GO term was found in common between the genes which were deregulated by zebularine in the dark and in the light, except for the “L-phenylalanine catabolic process”. Interestingly, this GO term was associated with the genes down-regulated by zebularine (Figure 7), suggesting that the phenylpropanoid pathway may be specifically down-regulated at the gene expression level in zebularine treated cells. This result was unexpected regarding the increase in anthocyanin accumulation in these two conditions, prompting a more thorough analysis of the gene deregulations associated with the phenylpropanoid and flavonoid biosynthesis pathways.

#### 3.2.1. Genes Involved in the Anthocyanin Pathway Are Differentially Regulated in Light and Dark Grown Cells

Transcriptomic data were used to evaluate the impact of zebularine on the expression of anthocyanin-related genes. For this purpose, all grape genes encoding enzymes and transcription factors potentially related to anthocyanin biosynthesis were recovered from the bibliography (Appendix A) and their expression was investigated using the RNA-seq data. A set of genes was also selected to perform qPCR experiments in order to confirm the RNA-seq data (Appendix A).

As expected, many genes encoding enzymes involved in anthocyanin synthesis, including *UFGT*, *MYBA1*, and genes coding for anthocyanin O-methyltransferases (AOMT, Vitvi01g01635; Vitvi01g02263; Vitvi01g02265), anthocyanin acyltransferases (AT, Vitvi03g01816; Vitvi03g00077), a glutathione S-transferase (GST, Vitvi04g00880), and anthocyanin transporters (Vitvi16g01913, Vitvi16g01210) were up-regulated in light-grown cells compared to those cultured in the dark (Figure 8A). The strongest effects were observed for the *UFGT* and *MYBA1* genes, with a 30- and a 42-fold increase in mRNA levels, respectively, in the light (Appendix A). Moreover, genes encoding a Phenylalanine Ammonia Lyase (PAL, Vitvi13g00622), a Cinnamate-4-Hydroxylase (C4H, Vitvi06g00803), and a 4-Coumarate-coA Ligase (4CL, Vitvi06g01318), which are critical enzymes of the general phenylpropanoid pathway, were also strongly up-regulated in the light (Figure 8A).

#### 3.2.2. Zebularine Treatment Enhances the Whole Flavonoid Biosynthesis Pathway in Illuminated Cells

The expression of the *UFGT* gene was enhanced in the presence of zebularine, although the mRNA level increased only by 1.8-fold, i.e., below the threshold chosen for DEG identification (Appendix A). A similar trend was observed for Vitvi01g02265 and Vitvi01g02263 coding for anthocyanin O-methyltransferases (AOMTs) and for Vitvi03g01816 coding for an anthocyanin 3-O-glucucoside-6″-O-acyltransferase (AT) (Appendix A). In addition, the expression of *AM3* (Vitvi16g01911), which encodes a vacuolar transporter for acetylated anthocyanins [15], increased over 5.5-fold in the presence of zebularine (Figure 8B and Appendix A). Hence, gene expression studies are consistent with both anthocyanin biosynthesis and transport into the vacuole being stimulated in the presence of zebularine in the light. Interestingly, the two genes, Vitvi06g01192 and Vitvi06g01885, coding for flavonoid 3′,5′-hydroxylases (F3’5’H), were also up-regulated by zebularine in the light (Figure 8B), nicely correlating with the increase in the proportion of tri-hydroxylated anthocyanins, which was associated with zebularine treatments (Figure 4 and Appendix A). In addition, genes coding for different enzymes associated with the general phenylpropanoid and flavonoid biosynthesis pathways (4-coumarate:CoA ligase (4CL), chalcone synthase (CHS), chalcone isomerases (CHI), flavanone 3-hydroxylase (F3H) and leucoanthocyanidin dioxygenase (LDOX/ANS)), were up-regulated in zebularine-treated cells (Figure 8B). More notably, genes specifically involved in the flavonol (flavonol synthase, FLS) and proanthocyanidins (anthocyanidin reductase, ANR) biosynthesis were also identified as zebularine-enhanced DEGs (Figure 6B). Finally, zebularine was shown to stimulate the expression of several genes coding for the transcriptional activators of the MYB family involved in the regulation of stilbene (MYB14, [89]), flavonol (MYBF1, [28,89]), and proanthocyanidin (MYBPA1, MYBPA2, MYBPAR, [26,29,31,33]) biosynthesis. Altogether these observations suggest that most probably not only anthocyanins but also stilbenes, flavonols and proanthocyanidins could be synthesized in higher quantities in cells treated with zebularine. Accordingly, stilbenes accumulated to higher levels in light grown zebularine-treated cells than in control cells treated with DMSO (Appendix A).

#### 3.2.3. Zebularine Specifically Induced Genes Associated with Anthocyanin Accumulation in Dark Grown Cells

In the dark, cells accumulated anthocyanin upon zebularine treatment only. Consistently, zebularine treatment was shown to induce *UFGT* and *MYBA1* expression (Figure 8C and Appendix A). Of note, *MYBA1* was not identified as a DEG when comparing zebularine-treated and control cells grown in the dark, because of the expression variability between replicates. However, the *MYBA1* gene was up-regulated in each of the three biological replicates, with an average four-fold increase in the zebularine-treated samples (Appendix A). In addition, genes coding for AOMT (Vitvi01g01635, Vitvi01g02263 and Vitvi01g02265), AT (Vitvi03g01816), GST (Vitvi04g00880) and MATE-type transporter (Vitvi16g01913) were also up-regulated by zebularine in dark grown cells (Figure 8C and Appendix A). However, a different MATE-type transporter encoding gene was induced by zebularine in the dark compared to the light. Whereas in the light, zebularine induced *AM3* (Vitvi16g01911), in the dark, it was *AM2* (Vitvi16g01913) (Appendix A). This could be related to the light-dependent expression of these two genes: *AM2* was barely expressed in the dark and highly expressed in the light, and the contrary was true for *AM3*. The F3’5’H encoding genes were expressed at very low levels in dark-grown cells and were not affected by the zebularine treatment. Accordingly, only di-hydroxylated anthocyanins (cyanidin and peonidin) were detected in the dark, whatever the treatment. In addition, three genes involved in the general phenylpropanoid pathway (Vitvi13g00622 (*PAL*), Vitvi06g00803 (*C4H*) and Vitvi02g00938 (*4CL*)), were up-regulated in dark grown zebularine-treated cells (Figure 8C). On the contrary the *MYBPA1* gene was repressed in zebularine-treated cells grown in the dark, suggesting a specific effect of zebularine on the anthocyanin biosynthetic pathway in these conditions, rather than a general effect on phenylpropanoid synthesis. Accordingly, there was no increase in stilbene content in dark grown cells treated with zebularine (Appendix A).

Altogether these results show that the genes which were up-regulated by zebularine in a highly significant manner (adjusted *p*-value < 0.05) were not the same in the dark and in the light (except for 4CL2 which is induced in both conditions but is expressed at very low levels). In particular, except for *UFGT* and *AM3*, anthocyanin-related genes are up-regulated by zebularine in the dark but not in the light. This observation may be related to the fact that these genes are poorly expressed in control dark grown cells, whereas they are highly expressed in control light grown cells, as shown in Appendix A. Therefore, any inducing effect of zebularine will inevitably be very limited in light grown cells. The most striking difference between the light and dark conditions is probably *MYBA1* behavior; its expression being induced by zebularine in the dark, whereas it was repressed by zebularine in the light.

### 3.3. Anthocyanin Accumulation in Zebularine-Treated Cells Correlates with a Slight Decrease in UFGT Methylation Status

In order to evaluate the impact of zebularine on the methylation status of GT cells, the global DNA methylation level was analyzed using HPLC analysis of methylated cytosines (mC). The global mC content of the DMSO-treated cells ranged from 8.06% in the dark to 9.08% in the light (Figure 9A), consistent with the mC content of Merlot grape berries at veraison, which was estimated at 7.77% by Whole Genome Bisulfite Sequencing [90]. Zebularine treatments were associated with a decrease from 9.08 to 8.2% in mC content in the light and no decrease in the dark (Figure 9A), suggesting that this drug was not very efficient in GT cell suspensions or, alternatively, that its hypomethylating effect was counterbalanced by an antagonistic process.

As methylation is mainly associated with repetitive sequences such as transposons (TEs), we analyzed the methylation status at selected TEs using McrBC-qPCR. The McrBC restriction enzyme only cuts methylated DNA. Therefore, the PCR efficiency is inversely correlated with the methylation level at a selected locus [42]. Two TEs were selected for this analysis: *GRET1* a gypsy-type retrotransposon ([91], accession: AB111100) and a DNA-type transposon of the CACTA family ([92], accession: AM487662). As shown in Figure 9B,C, the qPCR amplifications of digested CACTA_AM487662 and GRET1 TEs were, respectively, 1.9- and 1.5-fold higher, using DNA from zebularine-treated cells than of control cells, indicating that both TEs are less methylated after zebularine treatment in light grown cells. In dark grown cells, a similar trend was observed for CACTA_AM487662, with a 1.3-fold higher amplification in zebularine-treated samples compared to the controls (Figure 9B). In contrast, in dark grown cells, *GRET1* amplification was similar with and without zebularine treatment (Figure 9C). These results suggest that zebularine treatments were associated with a decrease in DNA methylation at specific loci, such as transposons, with a more important effect in the light than in the dark. Of note, other loci, such as *MYBA2* promoter behaved differently, with no significant variation in McrBC-qPCR amplification in the different conditions analyzed (Appendix A).

To determine whether the effect of zebularine on anthocyanin accumulation could be due to a change in the methylation status of *UFGT*, McrBC-qPCR assays were performed, focusing on its promoter regions. Interestingly, this quantitative analysis suggests that, indeed, the methylation level of *UFGT* was lower in zebularine-treated than in the mock-treated samples, in the region extending from −757 to −449 bp upstream from the transcription start (Figure 9D). The observed difference was significant for light grown cells; however, a high variability in the results obtained for the dark-grown DMSO-treated cells precludes any definitive statement concerning the dark grown cells.

### 3.4. Zebularine Induces a Stress Response in Grape Cells

#### Zebularine Slows down Cell Growth and Increases Cell Mortality

As already mentioned above, zebularine inhibits cell growth both in the light and the dark (Figure 2A,B). It also induces a decrease in cell viability (Appendix A). In total, 12 to 18% of the cells were identified as dead by trypan blue staining in 12 days old cultures in control (DMSO) conditions. This proportion tended to increase in zebularine-treated cells, varying for example between 21 and 25% after 7 days in the presence of zebularine at 75 µM (Appendix A). This observation suggests that zebularine negatively impacts cell viability. We further investigated a potential zebularine-induced stress effect by considering genes whose expression was affected by zebularine both in the light and in the dark but was not affected by light (Appendix A). A total of 144 genes were selected, among which the most highly induced by zebularine (log2 FC > 2) in light and dark conditions correspond to stress-related genes (Table 1): out of 26 genes with a functional annotation, 19 are related to the response of plants to various stresses.

An important group of genes is homologous to *A. thaliana* genes induced after a genotoxic stress [94]. Among them, *Vitvi04g01692*, which presents a 9.6- and 6.0-fold increase in expression in the presence of zebularine in light and dark, respectively, is homologous to *at1g19025*, a gene encoding a DNA repair metallo-beta-lactamase protein. *Vitvi05g01355*, which is homologous to atKU70, a key player in non-homologous end-joining pathway that repairs DNA double-strand breaks [94,95,96], is also induced in the presence of zebularine, to a lesser extent though (2.3- and 2.7-fold in light and dark, respectively). Similarly Vitvi13g01990 (12.1- and 3.4-fold increase in expression with zebularine in light and dark, respectively), is homologous to *AtSMR4*, which encodes a protein homologous to a SIAMESE/SIAMESE-RELATED (SIM/SMR) class of cyclin-dependent kinase inhibitors and is induced by DNA stress [94]. Finally, *Vitvi12g02472* which is the gene the most highly induced by zebularine in the dark (17.9-fold increase in expression) and is also highly induced in the light (11.8-fold), is homologous to at5g55490, which was considered by Yi et al. (2014) as a transcriptional hallmark of the DNA damage response, regardless of the type of DNA stress [94].

Additional genes induced in zebularine-treated cells are related to oxidative stress response. The *Vitvi12g00272* gene (7.4- and 6.9-fold induction in the light and dark, respectively) is homologous to at5g36160 coding for a tyrosine transaminase involved in tocopherol biosynthesis (Wang et al., 2019). Similarly, the *Vitvi14g00332* gene (5.4- and 5.2-fold induction in the light and dark, respectively) encodes a geranylgeranyl diphosphate reductase known to provide phytol for tocopherol synthesis (Tanaka et al., 1999). Furthermore, three genes induced by zebularine were also identified by Ramel et al. (2007), as genes up-regulated by atrazine, an herbicide which acts through induction of oxidative stress: *Vitvi09g00559* (Glyoxalase); *Vitvi01g01572* (AAA-ATPase) and *Vitvi08g00076* (efflux carrier) [93]. Several genes related to cell detoxification were identified, including two genes coding for glutathione S-transferases (Vitvi17g01381 and Vitvi17g00593) and two others for detoxification-related transporters (Vitvi08g01112 and Vitvi08g00076). Finally, seven other zebularine up-regulated genes were related to stress response (Table 1). For example, Vitvi17g01550 codes for a brassinosteroid-signaling kinase and Vitvi14g01439 for a retinoblastoma related protein, whose *Arabidopsis* closest homologous, RBR1, plays an important role in the detoxification response to DNA damage [94,95,96].

Altogether this transcriptional profile demonstrates that zebularine treatments, in the light and in the dark, trigger a stress response and suggests genomic lesions and oxidative injuries as already reported [60,61,62,63].

## 4. Discussion

The model used in this study, GT cell suspensions, which were originally established by Dr Pech in 1978 (ENSAT, Toulouse, France) from pulp fragments of young grape berries (as described in [97]) has been extensively used since then to study the production of anthocyanins and its regulation, since GT cell suspensions have the property to produce anthocyanins under continuous light (reviewed in [3,81]). However, these cells are highly heterogeneous, comprising mixtures of cells with different anthocyanin content [83,97,98,99]. Moreover, the non-pigmented cells are predominant, and their higher growth rate leads to a progressive reduction of the proportion of anthocyanin-producing cells, hence to a progressive loss of color of the cell suspension after recurrent subcultures [99]. Different groups have also reported a high level of variability in the capacity of GT cells to produce anthocyanins between subcultures and between flasks in the same subculture [80,83,99,100]. Finally different treatments were shown to positively affect anthocyanin production [81] in GT cell suspensions, including light [80], high sugar and low nitrogen concentrations [97,98,101,102], Pi deficiency [82], high osmotic potential [97], and different chemical effectors, including methyl jasmonate [103,104,105], jasmonic acid [80], and ABA [105,106].

To better understand the mechanisms underlying the regulation of anthocyanin accumulation in grape cells, we have investigated the role of DNA methylation in this process. The contribution of DNA methylation to the control of anthocyanin production would be consistent with the increasing number of data showing that DNA methylation plays a role in the regulation of the accumulation of secondary metabolites [42,107,108,109,110]. In particular, anthocyanin production is known to be under epigenetic control in other species such as apple, pear or radish [44,45,46,47]. The role of DNA methylation in anthocyanin production was analyzed in GT cells in light and dark conditions using a pharmacological approach to reduce genomic DNA methylation. Zebularine was chosen for this study because its hypomethylating effect was reported for different plant species [53,58,66,111]. Zebularine is a structural analog of methylcytosine (mC), which is incorporated into the DNA molecules where it can establish stable bounds with DMT and inactivate them [55,56].

### 4.1. Zebularine Has a Positive Impact on Anthocyanin Production Irrespective of the Growing Conditions and Mediates Limited Methylation Changes

Two cell suspensions were established from calli grown under continuous light. One was maintained in the same light condition as the calli, whereas the other was transferred to dark conditions. After four subcultures, dark grown cells lost their ability to produce anthocyanins. Zebularine treatments led to a dose-dependent stimulation of anthocyanin production in light grown cells and was sufficient to trigger anthocyanin production in non-pigmented dark-grown cells, as demonstrated by anthocyanin quantification 9 days after drug supplementation (Figure 4). It should be noted that zebularine did not affect all cells equivalently: only a fraction of cells became pigmented in the dark and the cell population remains highly heterogeneous in the light (Figure 3). Interestingly, the pigmented cells were often observed as clusters, suggesting that the competence for anthocyanin production may be heritable.

As expected, the global mC percentage was decreased in zebularine-treated light grown cells, although this effect was quite limited. In contrast, no significant change in the global genomic methylation level was observed in zebularine-treated dark grown cells (Figure 9). Accordingly, the methylation status of the GRET1 retrotransposon was reduced in the light only, whereas the analysis of a CACTA DNA transposon revealed a zebularine-dependent decrease in DNA methylation level in both conditions, more limited in the dark than in the light though. Such a limited impact on the global methylation level was surprising, considering the sharp decrease observed in tobacco BY2 cell suspension under very similar zebularine treatments [111]. This difference in zebularine consequences on DNA methylation may be related to the different genomic methylation levels of the two models: while the percentage of mC is close to 31% in tobacco BY2 cells [111], it was below 9.5% in GT grapevine cells. Alternatively, the limited impact of zebularine on the mC content of GT cells may be linked to specific properties of this model, which may not be able to efficiently metabolize this drug. Cells have to phosphorylate zebularine before it can be incorporated into DNA [93]. Possibly this phosphorylation could be more efficient in the light, explaining the differences observed between light and dark-grown cells. Finally, zebularine treatments led to a reduction in cell growth (Figure 2) and, therefore, to a limited synthesis of new DNA molecules, which in turn may contribute to mitigate the impact of zebularine on DNA methylation level.

### 4.2. Zebularine Induces a Stress-Response and Inhibits the Growth of GT Cell Suspensions

Zebularine not only inhibits cell growth but also induces cell death even at the lowest concentration used (25 µM) (Appendix A). These observations, together with the RNA-seq analyses, suggested that zebularine treatments were associated with a genotoxic stress. The most up-regulated genes after zebularine treatments include genes homologous to *AtSMR4* coding for a well characterized actor of the plant genomic DNA repair response [94] and *At1g19025*, belonging to the DNA repair metallo-beta-lactamase family. Moreover, many genes highly induced by zebularine have been shown to be regulated in response to toxic chemicals, such as atrazine [112], or by γ-rays [113,114,115,116] or by ROS accumulation [94]. This is consistent with recent reports characterizing zebularine as a genotoxic drug triggering both ATM- and ATR-dependent mechanisms and causing post-replicative cell cycle arrest [60,61,62,63]. Importantly anthocyanin accumulation has been repeatedly correlated to stress conditions and growth rate limitation both in cell cultures [3,80,81,117] and in intact plants [118]. Interestingly, γ-irradiations, which are known to induce ATM- and ATR-dependent responses [113] similar to zebularine, were also shown to induce anthocyanin accumulation [119,120], and the upregulation of genes linked to anthocyanin biosynthesis and transport [121]. Therefore, it cannot be ruled out that the enhancement of anthocyanin biosynthesis in the light and its induction in the dark may in part be linked to the stress response induced by zebularine, rather than to a direct demethylating effect of this drug. Nevertheless, we do not favor this stress hypothesis given the limited number of cells which acquire the capacity to produce anthocyanin upon zebularine treatment. We would expect a more general trend in the cell population for a non-specific stress response. 

### 4.3. The Expression Analysis of Anthocyanin-Related Genes Revealed Different Deregulation Patterns in the Dark and in the Light, Suggesting That the Mode of Action of Zebularine Could Be Different in These Two Situations 

As described above, zebularine differentially affects the global mC percentage depending on the dark/light conditions. In the same way, its effect on gene expression was different depending on the dark/light conditions (Figure 5A, compare Figure 6 and Figure 7). This could be partly linked to the high dissimilarity in the transcriptomes of the control cells depending on their growth condition (light or dark). More precisely, the identity of the genes differentially expressed between light and dark conditions in control cells suggests a divergence in their metabolic states, with control light grown cells presenting hallmarks of carbon starvation. Interestingly, metabolic analyses confirmed this observation. Control light grown cells were characterized by a lower content in soluble sugars (glucose + fructose) than all other cell samples and accumulated high amounts of asparagine (Appendix A), which was shown to be produced in response to carbon starvation [122,123,124,125,126]. Altogether these observations suggest that control light grown cells had deeply reshuffled their metabolism in response to carbon starvation when they were harvested. This was not the case for all other samples. Hence, control dark grown cells as well as zebularine-treated cells still contain sugars and do not accumulate high quantities of asparagine (Appendix A), suggesting that sugar consumption was slowed down in the dark and by the drug.

Another important physiological discrepancy between control light and dark grown cells is their different ability to produce anthocyanins. As a consequence, while the accumulation of anthocyanins in dark grown cells corresponds to the initiation of this biosynthetic pathway, in light grown cells the anthocyanins are already produced in the absence of any zebularine treatment, which essentially results in an increased accumulation of these compounds. This is well illustrated by the comparison of gene expression in DMSO control conditions. While in the dark, *UFGT* and *MYBA1* were barely expressed (9 and 5 RPKM, respectively), they were highly expressed in the light (256 and 188 RPKM, respectively) (Appendix A). Hence, the induction rate of the *UFGT* gene, following zebularine treatment, is by far higher in the dark (log2-fold induction of 3.6) than in the light (log2-fold induction of 0.6). In addition, the identity of the genes up-regulated by zebularine differs between light and dark grown cells. In the light, the most induced genes belong to the intermediary part of the flavonoid biosynthesis pathway. They include *CHS*, *CHI*, *F3’5’H*, *F3H*, and *LDOX* (Figure 8B), which likely reflects a diversion of part of the metabolic flux toward proanthocyanidins and flavonol biosynthesis, in addition to anthocyanin biosynthesis. The induction of stilbene accumulation in these cells supports this hypothesis (Appendix A), as well as the up-regulation of *ANR* and *MYBPA1* which are main regulators for PA-related gene expression [26]). In contrast, genes identified as up-regulated by zebularine in dark-cultured cells are located in the phenylpropanoid pathway (one *PAL*, one C4H and one 4CL gene) and in the lower part of the flavonoid biosynthesis pathway (*UFGT*, *AT*, *AOMT*) (Figure 8C). In this case, the C flux along the phenylpropanoid pathway is mainly directed toward anthocyanin production. Consistently, the stilbene content of dark-grown cells was not reproducibly increased by zebularine (Appendix A).

### 4.4. Zebularine-Dependent Activation of Anthocyanin Accumulation Was Correlated with an Up-Regulation of UFGT, Together with a Local Decrease in DNA Methylation in UFGT Promoter

Although it is described as a main regulator of the anthocyanin biosynthetic pathway, *MYBA1* expression was not reproducibly up-regulated by zebularine. In fact, it was induced in the dark but repressed in the light (Appendix A). In contrast, *UFGT* expression was induced by zebularine in light and dark, with a good correlation between *UFGT* expression level and anthocyanin cell content (Appendix A). Altogether these observations suggest that *MYBA1* may not be a determinant factor for the control of *UFGT* expression in grape GT cells in the presence of zebularine, as was already reported in the case of the *UFGT* gene induction by ethylene in GT suspension cells and in Cabernet Sauvignon berries [127].

Zebularine’s global effect on DNA methylation was shown to be limited in the light and undetectable in the dark (see above). Yet, this does not imply that the DNA methylation profile was not changed eventually resulting in changes in gene expression. Indeed, we observed a limited but consistent decrease in DNA methylation at the *UFGT* promoter (−757 to −449 bp upstream from the transcription start). As cell populations are highly heterogeneous and contain a defined amount of cells accumulating anthocyanins, therefore expressing the *UFGT* gene, it is consistent to observe only a limited reduction in *UFGT* DNA methylation levels. Whether the reduction in DNA methylation is causal to *UFGT* gene induction is so far unclear and would be consistent with the anthocyanin pathway being regulated by DNA methylation in various species, although different genes might be targeted depending on the species. Hence, whereas a *MYB* gene was implicated in pear, apple, and radish [44,45,46,47], and one *CHS* gene in orchids [51], the *UFGT* gene could be involved in grape GT cells. Because UFGT activity is a prerequisite for anthocyanin biosynthesis in grapes [25,128], the DNA methylation status at the *UFGT* promoter would work as an on/off switch for the production of these pigments in grape GT cells.

## 5. Conclusions

As a conclusion, zebularine treatment of grape cells results in complex and multiple effects including genotoxic and oxidative stresses, in addition to direct effects on DNA methylation. Hence, we cannot formally rule out that the enhancement or induction of anthocyanin accumulation upon zebularine treatment results from a combination of factors, which might also differ depending on the cell growing conditions. Among these mechanisms, anthocyanin accumulation in GT cells seems, at least in part, mediated through DNA demethylation at the *UFGT* gene. Whether a general DNA methylation dependent mechanism regulating anthocyanin accumulation in grape berries exists, remains, however, to be demonstrated.

## Figures and Tables

**Figure 1 genes-13-01256-f001:**
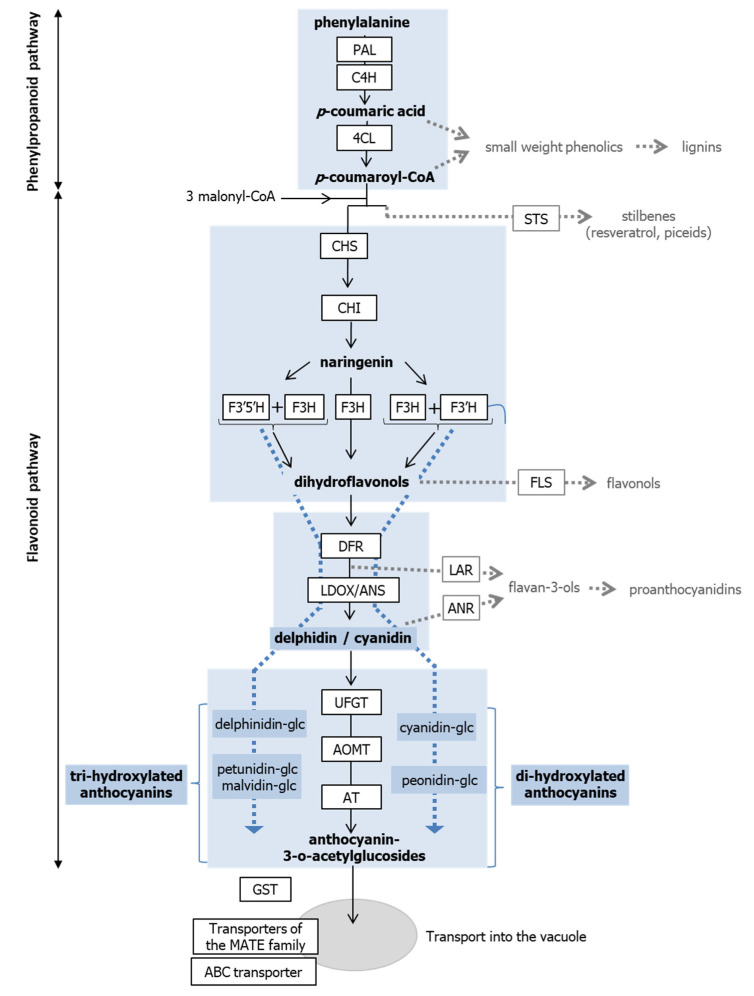
Anthocyanin biosynthesis pathway. Enzyme names are indicated in boxes. Enzyme names are as follows: PAL, phenylalanine ammonia lyase; C4H, cinnamate 4-hydroxylase; 4CL, 4-coumarate:CoA ligase; STS, stilbene synthese; CHS, chalcone synthase; CHI, chalcone isomerase; F3H, flavanone 3-hydroxylase; F3′H, flavonoid 3′-hydroxylase; F3′5′H, flavonoid 3′,5′-hydroxylase; FLS, flavonol synthase; DFR, dihydroflavonol 4-reductase; LAR, leucoanthocyanidin reductase; LDOX, leucoanthocyanidin dioxygenase; ANR, anthocyanidin reductase; UFGT, UDP glucose: flavonoid-3-*O*-glucosyltransferase; AOMT, anthocyanin *O*-methyltransferases; AT, anthocyanin 3-*O*-glucucoside-6″-*O*-acyltransferase; GST, glutathione S-transferase; MATE, multidrug and toxic extrusion, ABC transporters, transporters of the ATP binding cassette protein family. In parallel to the biosynthesis pathway, important anthocyanin intermediates are shown in blue boxes: delphidin, delphinidin-glc, petunidin-glc, and malvidin-glc correspond to tri-hydroxylated molecules whose synthesis involves F3′5′H and F3H. Cyanidin, cyanidin-glcm and peonidin-glc correspond to dihydroxylated molecules whose synthesis depends on F3′H and F3H.

**Figure 2 genes-13-01256-f002:**
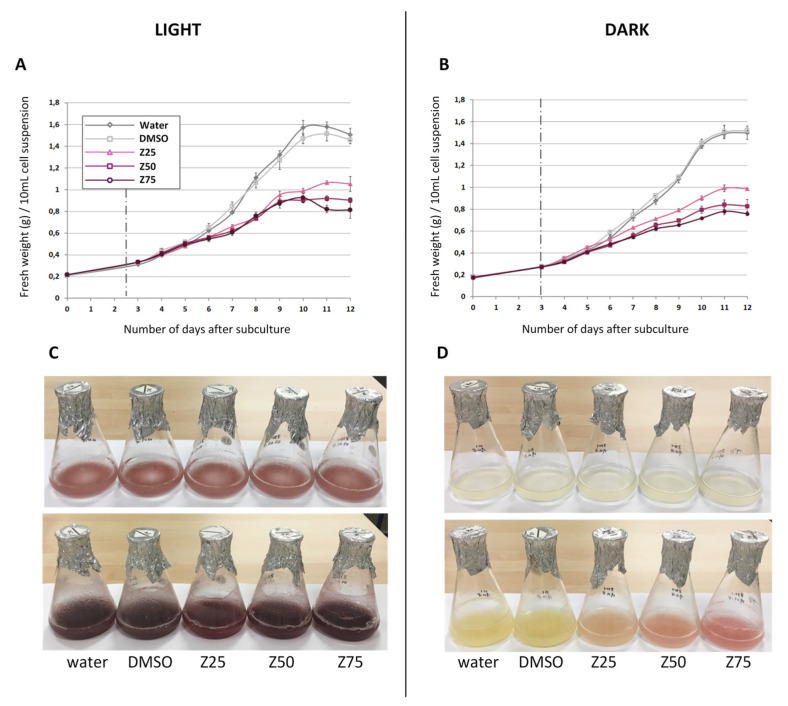
Zebularine affects GT cell growth and anthocyanin accumulation. Treatment with zebularine was performed 3 days after sub-culturing, in the light (**A**,**C**) and in the dark (**B**,**D**), using three concentrations 25 µM (Z25), 50 µM (Z50), and 75 µM (Z75). Two different controls were included (water and DMSO). Growth curves (**A**,**B**) were established by measuring the fresh weight of 10 mL samples after vacuum filtration each day after the sub-culture (day 0) until day 12. The values represent the means of three biological replicates and the bars the corresponding standard deviations. Photographs were taken at the first day (0 day) (upper pictures) and last day (12th day) (lower pictures) after sub-culturing (**C**,**D**).

**Figure 3 genes-13-01256-f003:**
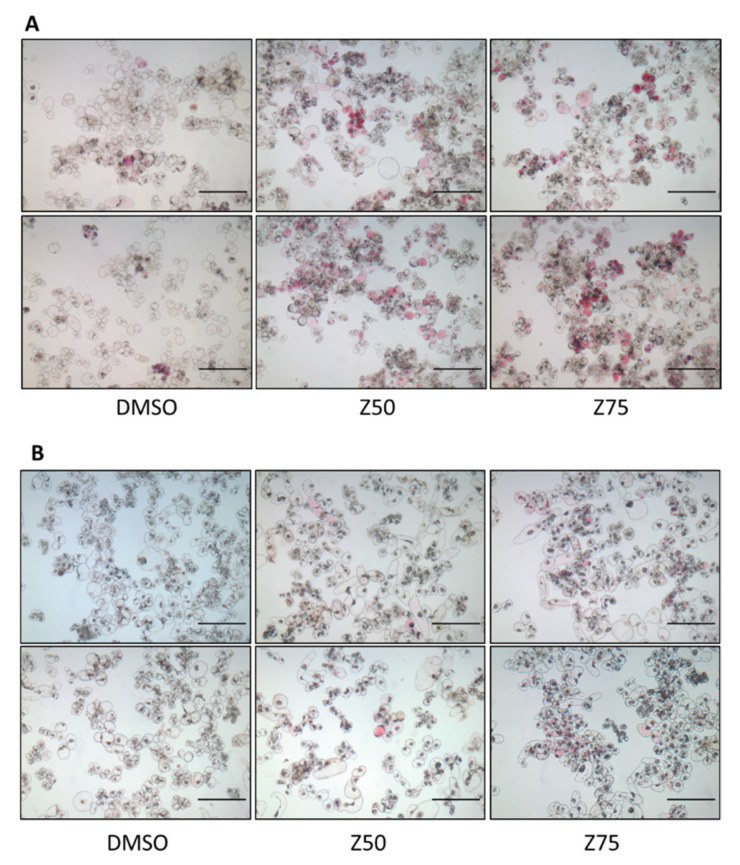
GT cell suspensions are heterogeneous: only a limited number of cells accumulate anthocyanins. Treatments with zebularine were performed 3 days after sub-culturing in the light (**A**) and the dark (**B**). Nine days after zebularine addition, cells were harvested, and microscopic observations were performed with an Axiophot Fluorescent microscope (Zeiss, Oberkochen, Germany) in brightfield mode. Bars = 250 µm.

**Figure 4 genes-13-01256-f004:**
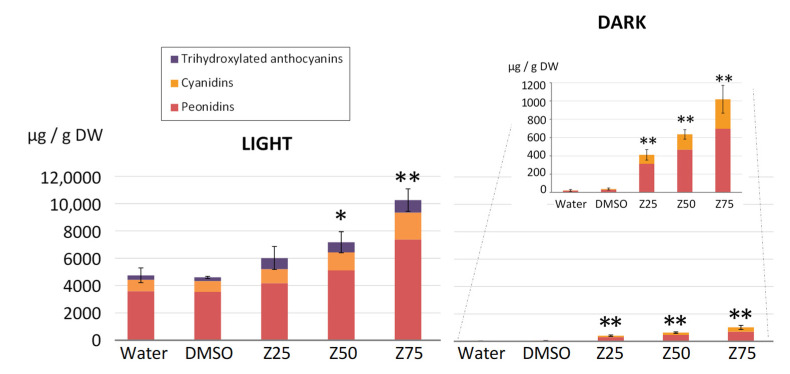
Analysis of zebularine impact on anthocyanin accumulation in the light and in the dark. Treatments with zebularine were performed 3 days after sub-culturing. Nine days after zebularine addition, cells were harvested, and anthocyanins were quantified. Cell total anthocyanin content was calculated as the sum of each individual anthocyanin. Values are the mean ± SD of three biological replicates. Asterisks indicate significant difference*s* in the total amounts of anthocyanins, as determined by a Welsh’s *t*-test *(n* = 3*)* based on the mean differences between zebularine-treated and DMSO-treated samples (* *p* < 0.05; ** *p* < 0.01). The quantities of peonidins, cyanidins, and trihydroxylated anthocyanins in each condition are indicated in the bars.

**Figure 5 genes-13-01256-f005:**
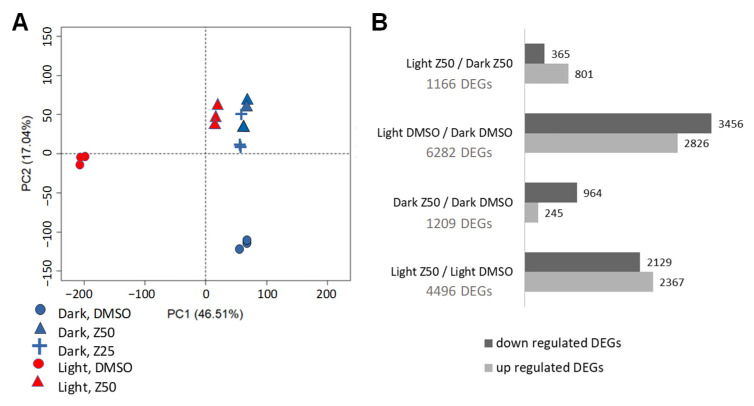
Transcriptome response to zebularine treatment in the light and in the dark. (**A**) Principal component analysis illustrating the relationships of the RNA-seq libraries generated from five different cell suspensions: light grown cells treated with DMSO or zebularine 50 µM and dark grown cells treated with DMSO or zebularine 25 or 50 µM. (**B**) Number of differentially expressed genes (DEGs) obtained by DESeq2 analysis for each cell growth condition except dark, Z25.

**Figure 6 genes-13-01256-f006:**
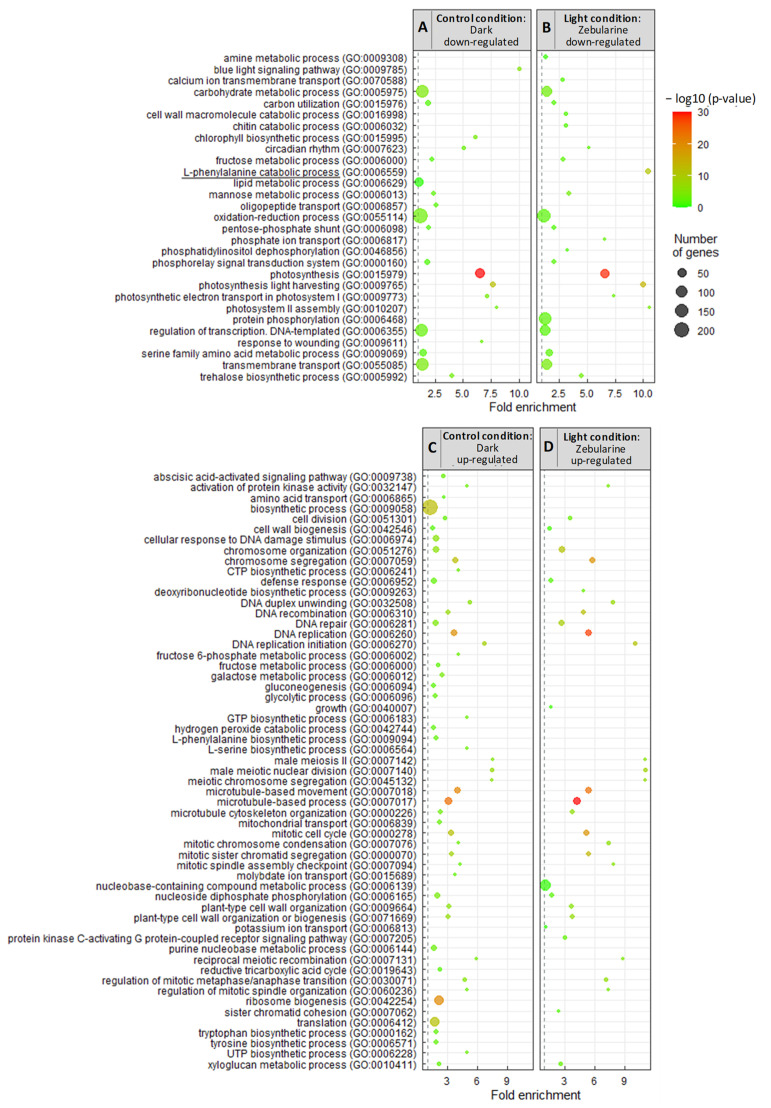
Identification of the differentially expressed gene (DEG) biological functions by a GO enrichment analysis revealed a high similarity between the genes which were deregulated by light in the control cells and the genes which were deregulated by zebularine in the light. The GO analysis (biological functions) was conducted for the genes which were down-regulated in the DD samples compared to the LD samples (**A**), down-regulated in the LZ50 samples compared to the LD samples (**B**), up-regulated in the DD samples compared to the LD samples (**C**), and up-regulated in the LZ50 samples compared to the LD samples (**D**). The common biological process between light- and dark- deregulated genes (comparison between Figure 6 and Figure 7) is underlined.

**Figure 7 genes-13-01256-f007:**
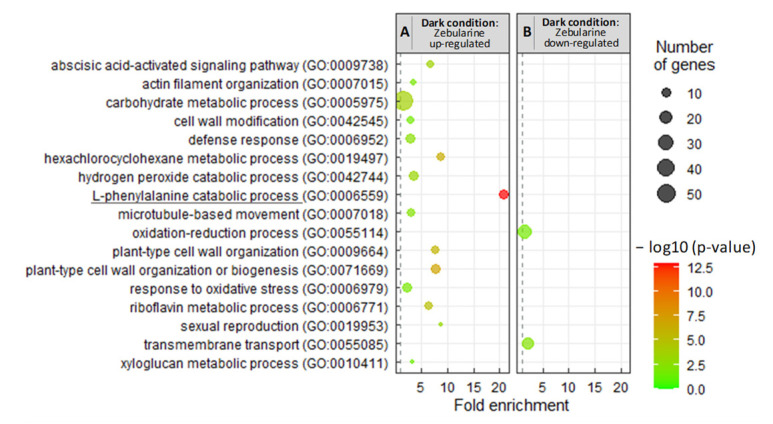
Analysis of the GO biological processes enrichment revealed few similarities between the genes which were deregulated by zebularine in the dark and in the light. The GO analysis (biological processes) was conducted for the genes which were up-regulated in the DZ50 samples compared to the DD samples (**A**) and down-regulated in the DZ50 samples compared to the DD samples (**B**). The common biological process between light- and dark- deregulated genes (comparison between Figure 6 and Figure 7) is underlined.

**Figure 8 genes-13-01256-f008:**
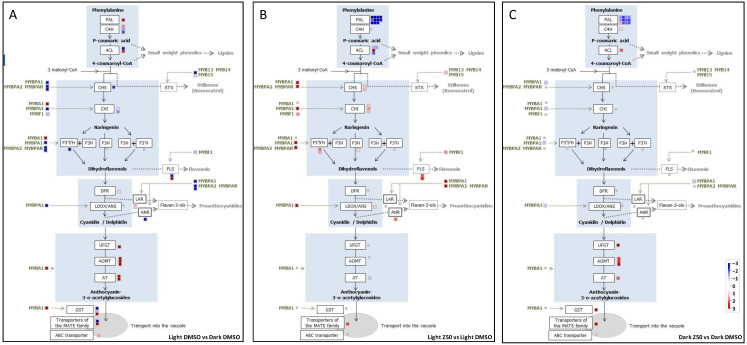
MAPMAN schematic representation of the anthocyanin biosynthesis pathway enlightening genes which were differentially expressed (|log2 FC| > 1, and padj > 0.05) (**A**) in the DMSO-treated light grown cells compared to the DMSO-treated dark grown cells, (**B**) in the zebularine-treated light grown cells compared to the DMSO-treated light grown cells, and (**C**) in the zebularine-treated dark grown cells compared to the DMSO-treated dark grown cells. (**A**) Red squares stand for genes up-regulated in light grown cells compared to dark grown cells and blue squares for genes down-regulated in light grown cells compared to dark grown cells. (**B**,**C**) Red squares stand for genes up-regulated in the presence of zebularine and blue squares for genes down-regulated by zebularine. The color intensity is proportional to the log2 FC as indicated in the upper left corners. The results obtained for the STS genes were not included since no specific behavior could be associated to this very large group of genes. Up-regulated genes in (**A**): PAL: Vitvi13g00622; C4H: Vitvi06g00803; 4CL: Vitvi06g01318; MYBA1: Vitvi02g01019; FLS: Vitvi18g02541; LAR: Vitvi17g00371; UFGT: Vitvi16g00156; AOMT: Vitvi01g01635, Vitvi01g02263, Vitvi01g02265; AT: Vitvi03g01816, Vitvi03g00077, GST: Vitvi04g00880; MATE transporter: Vitvi16g01913; ABC transporter: Vitvi16g01210; Down-regulated genes genes in (**A**): C4H: Vitvi11g01045; Vitvi11g00924; 4CL: Vitvi02g00938; MYB14: Vitvi07g00598; MYB15: Vitvi05g01733; MYBPA1: Vitvi15g00938; MYBPA2: Vitvi11g00099; MYBPAR: Vitvi11g00097; CHS: Vitvi14g01449; CHI: Vitvi13g01911; Vitvi04g00175; MYBF1: Vitvi07g00393; F3’5’H: Vitvi08g01637; FLS: Vitvi18g02538; DFR: Vitvi18g00988; LAR: Vitvi01g00234; ANR: Vitvi10g02185; GST: Vitvi19g01328; Vitvi19g01338; MATE transporter: Vitvi16g01911; Up-regulated genes in (**B**): 4CL: Vitvi02g00938; CHS: Vitvi14g01448, Vitvi14g01449; CHI: Vitvi13g01911, Vitvi14g01683, Vitvi04g00175; F3H: Vitvi04g01454; F3’5’H: Vitvi06g01885, Vitvi06g01192; LDOX/ANS: Vitvi02g00435; MYB14: Vitvi07g00598; MYBF1: Vitvi07g00393; FLS: Vitvi18g02538, Vitvi18g02541; ANR: Vitvi10g02185; MYBPA1: Vitvi15g00938; MYBPA2: Vitvi11g00099; MYBPAR: Vitvi11g00097; MATE transporter: Vitvi16g01911; Down-regulated genes in (**B**): PAL: Vitvi00g01367; Vitvi16g01507; Vitvi16g00061; Vitvi16g00055; Vitvi08g01022; Vitvi16g01502; Vitvi16g01503; Vitvi16g00057; Vitvi16g00066; Vitvi16g00054; Vitvi16g00060; 4CL: Vitvi11g01257; Vitvi18g00126; Vitvi06g01318; AT: Vitvi03g00077; Up-regulated genes in (**C**): PAL: Vitvi13g00622; C4H: Vitvi06g00803; 4CL: Vitvi02g00938; UFGT: Vitvi16g00156; AOMT: Vitvi01g01635, Vitvi01g02263, Vitvi01g02265; AT: Vitvi03g01816; GST:Vitvi04g00880; MATE transporter: Vitvi16g01913; Down-regulated genes in (**C**): PAL: Vitvi00g01367; Vitvi16g00066; Vitvi16g00054, Vitvi16g00060, Vitvi08g01022, Vitvi16g01507, Vitvi16g00061, Vitvi16g00055, Vitvi16g01502, Vitvi16g01503, Vitvi16g00057; AT: Vitvi03g00077; MYBPA1: Vitvi15g00938.

**Figure 9 genes-13-01256-f009:**
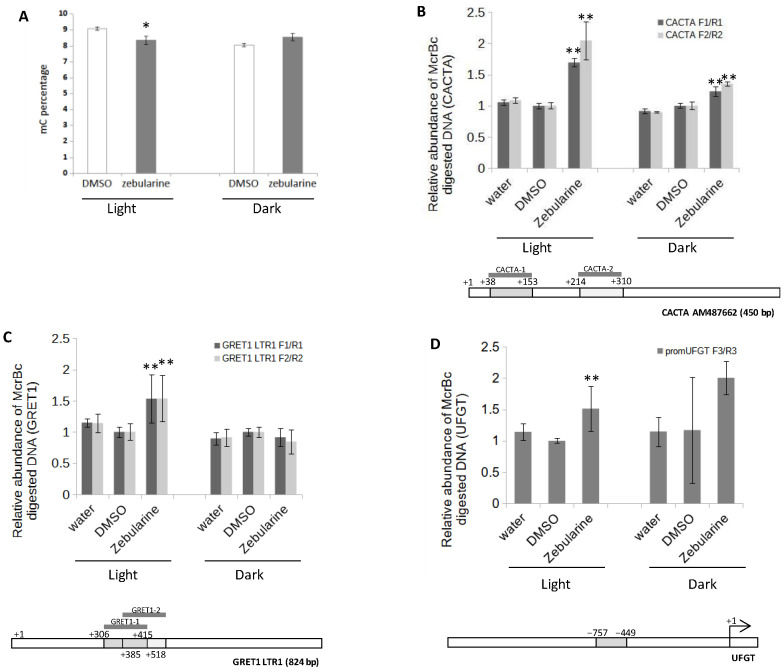
Zebularine hypomethylating effect is light- and locus-dependent. Global DNA methylation percentages (**A**) and locus-specific methylation levels (**B**–**D**) were evaluated. (**A**) Bars represent the mean value of three to five biological replicates with their standard deviations. Three biological replicates were used for all DMSO-treated samples, five biological replicates for the light grown zebularine-treated cells (three replicates treated with 25 µM and two with 50 µM), and three biological replicates for the dark grown zebularine-treated cells (two replicates treated with 25 µM and one with 50 µM). (**B**–**D**) Locus-specific methylation levels were measured by McrBC-qPCR as described in the material and method. Since the McrBC is a methylation-dependent restriction enzyme, the relative abundance of the gDNA sequences after the McrBC digestion is inversely proportional to its methylation level. Four different sequences were analyzed: two sequences corresponding to the transposon AM487662 of the CATAVINE1 family (**B**), a sequence from the GRET1 transposon (accession AB111100) (**C**), and a sequence located in *UFGT* 5’region, 449 bp upstream from the ATG (**D**). The position of the primers used for the different qPCR is shown below the qPCR quantification. DNA input amounts in the different samples were normalized by amplifying a sequence that was not cut by the enzyme (no methylation site), i.e., actin. Graphs show the relative abundance of the amplicons in the different McrBC-treated samples (normalization to ACTIN and to DMSO-treated samples). Bars represent the mean values for three to seven biological replicates with their standard deviations. Three biological replicates were used for all water- and DMSO-treated cells, five for the dark grown zebularine-treated cells (two replicates treated with 25 µM, one with 50 µM and one with 75 µM) and seven for the light grown zebularine-treated cells (three replicates treated with 25 µM, two with 50 µM and two with 75 µM). Stars indicate significance in a Welch *t*-test based on the mean difference between zebularine-treated and DMSO-treated samples (* *p* < 0.05; ** *p* < 0.01). The McrBC-qPCR experiment was performed twice with reproducible results.

**Table 1 genes-13-01256-t001:** Genes up-regulated by zebularine both in light and dark grown cells. A number of 68 DEGS were identified as specifically up-regulated by zebularine in the light and in the dark with a log2-fold change (log2 FC) > 1 and an adjusted *p*-value (*p* adj) > 0.05. Among them, 38 genes were associated with a log2 FC > 2 at least in one comparison (DL versus DZ50 or LD versus LZ50). Finally, a putative function could be assigned for 26 of these 38 genes, by comparison with homologous genes found in *Arabidopsis thaliana* or *Nicotiana tabacum* (*N. tabacum*) genomes. The orange color indicates genes related to genotoxic stress and/or DNA repair processes. The blue color indicates stress-related genes. * indicated *Arabidopsis thaliana* genes induced by atrazine [93].

Gene Identifier	Dark		Light		Function	Homologous Gene(s)
	log2FC	*p* adj	log2FC	*p* adj		
*Vitvi12g02472*	4.2	3 × 10^−79^	3.6	8 × 10^−39^	GEX1 (unknown function)	at5g55490
*Vitvi13g01990*	1.8	2 × 10^−3^	3.6	8 × 10^−18^	Cyclin-dependent protein kinase inhibitor	at5g02220 (SMR4)
*Vitvi04g01692*	2.6	3 × 10^−49^	3.3	1 × 10^−74^	DNA repair	at1g19025
*Vitvi17g01550*	3.3	2 × 10^−17^	4.3	6 × 10^−12^	Brassinosteroid-signaling kinase	at5g59010 (ATBSK5)
*Vitvi19g02101*	2.9	1 × 10^−9^	4.2	4 × 10^−8^	Ferredoxin-fold anticodon-binding domain protein	at1g55790
*Vitvi17g00593*	2.3	1 × 10^−2^	4.1	2 × 10^−3^	Glutathione S transferase	at1g74590 (GSTU10)
*Vitvi17g01381*	3.8	2 × 10^−27^	3.2	1 × 10^−19^	Glutathione S transferase	at2g29420
*Vitvi08g01112*	3.2	4 × 10^−5^	2.7	4 × 10^−5^	ABC transporters and multidrug resistance system	at2g37360; at3g53510
*Vitvi12g00272*	2.8	5 × 10^−34^	2.9	1 × 10^−27^	Tyrosine transaminase	at5g36160
*Vitvi01g01572*	2.8	2 × 10^−2^	2.4	3 × 10^−2^	AAA-ATPase	at3g50940; at2g18193 *
*Vitvi14g00163*	2.5	3 × 10^−8^	2.6	6 × 10^−9^	Heavy metal-associated isoprenylated protein	at5g27690
*Vitvi08g00076*	2.4	5 × 10^−16^	1.8	2 × 10^−6^	Detoxification efflux carrier	at1g33110 *
*Vitvi14g00332*	2.4	1 × 10^−2^	2.4	4 × 10^−5^	Geranylgeranyl diphosphate reductase	Q9ZS34 (*N. tabacum*)
*Vitvi05g02234*	1.5	3 × 10^−2^	2.2	5 × 10^−4^	Disease resistance RPP8-like protein	at5g35450 (RPP8L3)
*Vitvi03g01650*	1.3	5 × 10^−3^	2.1	4 × 10^−20^	Pathogenesis-related protein	at2g14580
*Vitvi03g01542*	1.9	3 × 10^−8^	2.1	3 × 10^−19^	2-oxoglutarate and Fe-dependent oxygenase	at3g19000
*Vitvi09g00559*	1.3	5 × 10^−3^	2.1	3 × 10^−10^	Glyoxalase I family protein	at1g80160 *
*Vitvi02g01446*	1.5	2 × 10^−12^	2.1	1 × 10^−7^	Heat shock protein	at4g25200 (HSP23.6)
*Vitvi14g01439*	2.0	2 × 10^−16^	1.7	1 × 10^−12^	Retinoblastoma related protein	at3g12280 (RBR1)
*Vitvi09g00768*	4.0	2 × 10^−24^	6.9	1 × 10^−19^	Ubiquitin E3 SCF FBOX	at5g07610
*Vitvi12g00255*	2.0	2 × 10^−5^	3.2	5 × 10^−9^	NAC transcription factor	at4g28500
*Vitvi12g01880*	3.0	2 × 10^−63^	3.0	2 × 10^−70^	Cupin (storage protein)	at1g07750
*Vitvi10g02406*	2.2	3 × 10^−2^	3.0	3 × 10^−2^	MYB domain transcription factor	at2g02060
*Vitvi05g00582*	2.2	1 × 10^−38^	1.8	9 × 10^−17^	Calcium transporting ATPase	at3g22910
*Vitvi18g01607*	1.9	5 × 10^−5^	2.1	6 × 10^−6^	Protein kinase	at1g54610; at5g50860
*Vitvi06g00621*	1.6	8 × 10^−4^	2.0	3 × 10^−13^	UDP-glycosyltransferase	at1g07250 and homologous genes

## Data Availability

The RNA-seq data have been deposited in the European Nucleotide Archive (ENA) at EMBL-EBI under accession number PRJEB54585 (https://www.ebi.ac.uk/ena/browser/view/PRJEB54585).

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
