# Peer review of "Zebularine, a DNA Methylation Inhibitor, Activates Anthocyanin Accumulation in Grapevine Cells"

_genes, 2022, doi:10.3390/genes13071256_

Round 1

Reviewer 1 Report

DNA methylation plays an important role in plant development. This study elucidated the the role of the zebularine(a DNA methylation inhibitor) in anthocyanin accumulation in grape. These results are helpful to better know the effect of DNA methylation in fruit ripening of grapevine. But, there existed figure order, result analysis problems. So, the article can be accepted after minor revise.

1. Previous studies showed procyaniidin or anthocyanin from seed of grape, sweet potato or cranberry caould inhibit cancer cell growth. So, I cannot make sure the role in inhibition of cell growth is Zebularine or anthocyanin?

2. In Figure 3 and 4, the result of order should be consistent. First is dark treatment, second light treatment.

3. In this study, Zebularine promoted the accumulation of resveratrol in light condition. But in transcriptome analysis, only MYB14 gene was mentioned. So many stilbene genes were not identified? Moreover, these results were not related to this study.

Author Response

Point 1: Previous studies showed procyaniidin or anthocyanin from seed of grape, sweet potato or cranberry caould inhibit cancer cell growth. So, I cannot make sure the role in inhibition of cell growth is Zebularine or anthocyanin?

Response 1: It is true that both growth inhibition and anthocyanin accumulation are correlated to zebularine treatments. However, we believe that the growth inhibition is linked to a direct effect of zebularine rather than to an effect of anthocyanins. First of all, the anthocyanins produced in the GT cells are not solubilized in the cell medium but sequestrated in the vacuole (Conn et al 2010, see also figure 3). Therefore, they cannot exert any negative effect on cell physiology. Indeed, the inhibition of cancer cell growth by anthocyanins was observed when this molecule was supplied to the growth medium. Second, there is no correlation between cell anthocyanin content and cell growth: the GT control cell accumulate anthocyanin in the light, but not in the dark, however light grown cells do no grow slower that dark grown-cells. In addition, it was noted that anthocyanins accumulate in grapevine cell cultures after methyl jasmonate in combination or not with carbohydrates without growth alteration (Belhadj et al., 2008). Finally, a direct effect of zebularine on cell growth is consistent with results reported in several other studies: Baubec et al. 2009, 2015, Boonjing et al., 2020, Finnegan et al., 2018, Iwase et al., 2010, Malik et al, 2012, Liu et al., 2015, Nowicka et al., 2020.

Point 2: In Figure 3 and 4, the result of order should be consistent. First is dark treatment, second light treatment.

Response 2: Figure 2 and 3 has been modified accordingly: The presentation is now consistent between Figure 2, Figure 3 and Figure 4: the panel corresponding to the light condition is on the left of the figure, or on top of the figure. We decided to present first the light condition, and second the dark condition in order to fit with the result description in the text.

Point 3: In this study, Zebularine promoted the accumulation of resveratrol in light condition. But in transcriptome analysis, only MYB14 gene was mentioned. So many stilbene genes were not identified? Moreover, these results were not related to this study.

Response 3: Our analysis aims to analyze the effect of zebularine on the production of anthocyanin production. Stilbene production was not our focus. However, we found interesting to consider the metabolic flux diversion toward stilbene, flavonol and proanthocyanidin biosynthesis for a better comprehension of the variation in anthocyanin quantities upon zebularine treatments. Therefore, we have included the analysis of different genes related to stilbene, flavonol and proanthocyanidin biosynthesis (MYB13, MYB14, MYB15, FLS, MYBF1, LAR, ANR, MYBPA1, MYBPA2, and MYBPAR). We have not included the STS genes in this analysis for several reasons: (1) Forty-eight STS encoding genes have been identified in the grapevine genome, with at least 32 putatively functional ones (Parage et al, 2011, Vannozzi et al, 2012). (2) A few of these STS genes were detected among the DEGs identified in the frame of our analysis, but they did not exhibit a common behavior upon zebularine treatments. Hence their analysis did not provide any clue regarding the effect of zebularine on the metabolic flux toward stilbene synthesis. For these reasons, and also because we wanted to keep the figure 8 as simple as possible we have not presented the STS gene expression in figure 8. In order to clarify this point, a sentence has been added in the legend of figure 8: “The results obtained for the STS genes were not included since no specific behavior could be associated to this very large group of genes.”. In addition, a sentence was added in the legend of Figure S8 to specify that resveratrol could not be precisely quantified, due to its low abundance. In the study done by Belhadj et al. (2008), the authors also noticed that in cells, only trans- and cis-piceids can be detected and not resveratrol. However, resveratrol can be accumulated in the culture medium. Piceids are a storage form of stilbenes, this explains their stronger presence in cells. Resveratrol, as an active grapevine anti-microbial compound, could be preferentially released by the cells to fight potential pathogens.

Baubec, T.; Pecinka, A.; Rozhon, W.; Mittelsten Scheid, O. Effective, Homogeneous and Transient Interference with Cytosine Methylation in Plant Genomic DNA by Zebularine. Plant J. 2009, 57, 542–554, doi:10.1111/j.1365-313X.2008.03699.x.

Baubec, T.; Pecinka, A. Repair of DNA Damage Induced by the Cytidine Analog Zebularine Requires ATR and ATM in Arabidopsis. Plant Cell 2015, 27, 1788–1800, doi:10.1105/tpc.114.135467

Boonjing, P.; Masuta, Y.; Nozawa, K.; Kato, A.; Ito, H. The Effect of Zebularine on the Heat-Activated Retrotransposon ONSEN in Arabidopsis Thaliana and Vigna Angularis. Genes Genet. Syst. 2020, 95, 165–172, doi:10.1266/ggs.19-00046

Conn, S.; Franco, C.; Zhang, W. Characterization of Anthocyanic Vacuolar Inclusions in Vitis Vinifera L. Cell Suspension Cultures. Planta 2010, 231, 1343–1360, doi:10.1007/s00425-010-1139-4.

Finnegan, E.J.; Ford, B.; Wallace, X.; Pettolino, F.; Griffin, P.T.; Schmitz, R.J.; Zhang, P.; Barrero, J.M.; Hayden, M.J.; Boden, S.A.; et al. Zebularine Treatment Is Associated with Deletion of FT-B1 Leading to an Increase in Spikelet Number in Bread Wheat. Plant Cell Environ. 2018, 41, 1346–1360, doi:10.1111/pce.13164.

Iwase, Y.; Shiraya, T.; Takeno, K. Flowering and Dwarfism Induced by DNA Demethylation in Pharbitis Nil. Physiol. Plant. 2010, 139, 118–127, doi:10.1111/j.1399-3054.2010.01345.x.

Malik, G.; Dangwal, M.; Kapoor, S.; Kapoor, M. Role of DNA Methylation in Growth and Differentiation in Physcomitrella Patens and Characterization of Cytosine DNA Methyltransferases. FEBS J. 2012, 279, 4081–4094, doi:10.1111/febs.12002.

Liu, C.-H.; Finke, A.; Díaz, M.; Rozhon, W.; Poppenberger, B.; Baubec, T.; Pecinka, A. Repair of DNA Damage Induced by the Cytidine Analog Zebularine Requires ATR and ATM in Arabidopsis. Plant Cell 2015, 27, 1788–1800, doi:10.1105/tpc.114.135467.

Nowicka, A.; Tokarz, B.; Zwyrtková, J.; Dvořák Tomaštíková, E.; Procházková, K.; Ercan, U.; Finke, A.; Rozhon, W.; Poppenberger, B.; Otmar, M.; et al. Comparative Analysis of Epigenetic Inhibitors Reveals Different Degrees of Interference with Transcriptional Gene Silencing and Induction of DNA Damage. Plant J. 2020, 102, 68–84, doi:10.1111/tpj.14612.

Reviewer 2 Report

change 214  from DMT to DNA methyltransferase (DMT)

change 346 from (Figure 2AB) to (Figure 2A-B)

Author Response

Point 1: change 214  from DMT to DNA methyltransferase (DMT)

Response 1: The change has been made as requested

Point 2: change 346 from (Figure 2AB) to (Figure 2A-B)

Response 2: The change has been made as requested

Reviewer 3 Report

This paper deals with Zebularine, a DNA methylation inhibitor, that activates anthocyanin accumulation in grapevine cells. To better understand the mechanisms underlying the regulation of anthocyanin accumulation in grape cells, the authors have investigated the role of DNA methylation in this process. The experiments have been good designed and carried out. The methods used in this study are specific and adequate.

Abstract. please change “glucose:flavonoid-3-O-glucosyltransferase” to “glucose:flavonoid-3-O- glucosyltransferase”

Line 48 and 50: “p-coumaroyl- CoA” please change to “p-coumaroyl- CoA”

Line 58: “glucose:flavonoid-3-O-glucosyltransferase.” Please change to “glucose:flavonoid-3-O-glucosyltransferase.”

Figure 1: “: flavonoid-3-0-glucosyltransferase. “ Please change to “: flavonoid-3-O-glucosyltrans-ferase “ and “anthocyanin O-methyltransferases” please change to “anthocyanin O-methyltransferases” and “anthocyanin 3-O-glucucoside-6’’-O-acyltransferase” please change to “anthocyanin 3-O-glucucoside- 6’’-O-acyltransferase”.

Line 209: This reference is not in number “(Decendit et al., 1996)”

Author Response

Point 1: Abstract. please change “glucose:flavonoid-3-O-glucosyltransferase” to “glucose:flavonoid-3-O- glucosyltransferase”

Response 1: The change has been made as requested

Point 2: Line 48 and 50: “p-coumaroyl- CoA” please change to “p-coumaroyl- CoA”

Response 2: The change has been made as requested

Point 3: Line 58: “glucose:flavonoid-3-O-glucosyltransferase.” Please change to “glucose:flavonoid-3-O-glucosyltransferase.”

Response 3: The change has been made as requested

Point 4: Figure 1: “: flavonoid-3-0-glucosyltransferase. “ Please change to “: flavonoid-3-O-glucosyltrans-ferase “ and “anthocyanin O-methyltransferases” please change to “anthocyanin O-methyltransferases” and “anthocyanin 3-O-glucucoside-6’’-O-acyltransferase” please change to “anthocyanin 3-O-glucucoside- 6’’-O-acyltransferase”.

Response 4: The change has been made as requested

Point 5: Line 209: This reference is not in number “(Decendit et al., 1996)”

Response 5: The change has been made as requested
